# Marmosets mutually compensate for differences in rhythms when coordinating vigilance

Nikhil Phaniraj[1,2☯]*, Rahel K. Brügger[1☯], Judith M. Burkart[1,2,3]

**1** Institute of Evolutionary Anthropology, University of Zurich, Zurich, Switzerland, **2** Neuroscience Center Zurich, University of Zurich and ETH Zurich, Zurich, Switzerland, **3** Center for the Interdisciplinary Study of Language Evolution (ISLE), University of Zurich, Zurich, Switzerland

☯ These authors contributed equally to this work.
* rajpkn1@gmail.com

**Data Availability Statement:** All data and codes can accessed at https://doi.org/10.5281/zenodo.11084748.

**Funding:** This work was supported by the H2020 European Research Council (grant agreement No

## Abstract

Synchronization is widespread in animals, and studies have often emphasized how this seemingly complex phenomenon can emerge from very simple rules. However, the amount of flexibility and control that animals might have over synchronization properties, such as the strength of coupling, remains underexplored. Here, we studied how pairs of marmoset monkeys coordinated vigilance while feeding. By modeling them as coupled oscillators, we noted that (1) individual marmosets do not show perfect periodicity in vigilance behaviors, (2) nevertheless, marmoset pairs started to take turns being vigilant over time, a case of anti-phase synchrony, (3) marmosets could couple flexibly; the coupling strength varied with every new joint feeding bout, and (4) marmosets could control the coupling strength; dyads showed increased coupling if they began in a more desynchronized state. Such flexibility and control over synchronization require more than simple interaction rules. Minimally, animals must estimate the current degree of asynchrony and adjust their behavior accordingly. Moreover, the fact that each marmoset is inherently non-periodic adds to the cognitive demand. Overall, our study provides a mathematical framework to investigate the cognitive demands involved in coordinating behaviors in animals, regardless of whether individual behaviors are rhythmic or not.

## Author summary

Research suggests that synchronized animal behaviors often emerge from simple interaction rules. Mathematical models have been instrumental in revealing these underlying rules. Here, we employed mathematical modeling to study how marmoset monkeys coordinate vigilance and feeding behaviors in a situation where doing both actions simultaneously is not possible. We found that pairs of marmosets progress to a state where they show opposite behaviors, i.e., when one individual is feeding, the other is vigilant, and vice-versa. In order to achieve such coordinated state, the individuals must influence each other's behaviors, i.e., couple. We found that marmosets can couple flexibly and that they

101001295 to JMB); the National Centre for Competence in Research Evolving Language (agreement no. 51NF40_180888 to JMB); the Schweizerischer Nationalfonds zur Förderung der Wissenschaftlichen Forschung (award no. 31003A_172979 to JMB) as well as the Janggen-Pöhn-Stiftung (to RKB). The funders had no role in study design, data collection and analysis, decision to publish, or preparation of the manuscript.

**Competing interests:** The authors have declared that no competing interests exist.

couple more strongly if they are initially out-of-sync with their partner. Such ability to detect the current state of synchrony and adapt behavior accordingly is cognitively demanding. Our research thus demostrates that animals with more complex cognitive abilities can do much more than following simple interaction rules to synchronize with other individuals. Overall, our research (1) establishes marmosets as a strong candidate species for studying the cognitive aspects of social timing, (2) provides a novel mathematical framework that is tailored for studying synchronization in biological systems, and (3) underlines the implications of synchrony for marmosets and other animals.

## Introduction

Rhythmic phenomena involving multiple animals are widespread in nature [1–3]. From synchronized flashings of fireflies [4,5], call synchronization in frogs [6,7] to coupled neuronal activity in bats [8] or synchronized heart rates in humans [9]–synchronicity can be found in a huge variety of organisms in different behavioral contexts, at varying levels and serving multiple functions. Synchronicity not only includes patterns where individuals behave in the same way at the same time, as in the impressive synchronous light flashes of fireflies or coordinated movements of starling flocks [10]. Many phenomena show anti-phase synchronization in which individuals alternate behavior [11], as in vocal turn-taking in marmoset monkeys [12,13], meerkats [14], elephants [15] and plain-tailed wrens [16] or gestural exchanges of mother-infant dyads in chimpanzees and bonobos [17]. Whereas in-phase and anti-phase synchrony are the two extremes and frequently occur in nature, synchronization can, in principle, occur at any phase lag (e.g., [18]). Throughout this paper, we use 'synchronization' to refer to the phenomena of individuals showing repetitive behavior at the same rate, with any phase difference between them.

However, the complexity of these phenomena does not necessarily imply complex cognitive mechanisms. There is extensive evidence from invertebrates that synchronized patterns can emerge as epiphenomena of animals following very simple rules: fireflies, for example, simply flash sooner than usual whenever a neighbor flashes and the resulting synchrony emerges from these small changes in local interactions [19–21]. In such cases, where individuals are not synchronized to begin with, individuals need to interact in such a way that they influence the behavior of each other, i.e., 'couple', to reach a synchronized state. These simple mechanistic rules predict patterns of synchronization that are quite uniform, with minimal variation of rhythmic rates between individuals that are coupled and across behavioral bouts. Animals are also expected to show a limited range of possible adjustments to their inherent rhythm, making it difficult to adjust to more dissimilar stimulus rhythms [3]. This is especially important when looking at contexts such as gait synchronization [22,23] since the skeletal and motor system inherently limits the movement frequencies of animals [3]. When looking at patterns of behavior such as gestural exchanges these limiting factors to the flexibility of rhythmic rates might be less constrained but in turn synchronization might be more difficult to achieve.

What remains underexplored is how flexible synchronization patterns are in non-human animals with more complex cognitive abilities compared to invertebrates. Most studies investigating cognitive mechanisms underlying rhythmic behaviors focus on the perception and production of rhythmic patterns [24]. For example, typical rhythm perception tasks are performed as go/no-go tasks where animals are trained to respond to regular (isochronous) rhythms but not irregular (anisochronous) ones. The same previously trained animals are then confronted with novel regular and, importantly, irregular sequences ("anisochrony detection").

Individuals are expected to generalize the patterns of regularity to the novel sequences, i.e., they should again respond to regular sequences but not to irregular ones. Both rats [25] and starlings [26,27] are capable of solving such tasks (but not zebra finches: [28] and pigeons: [29]). Other studies use approach behavior (i.e., moving closer to one of two sound sources) to discern whether animals can discriminate between rhythmic patterns. These tasks seem especially promising when used with mating calls of frogs or insects, where animals seem to show species-specific rhythmic preferences [30,31]. Rhythm production studies in vertebrate animal models rely heavily on training (and rewarding anticipatory movements) and a laboratory setting (i.e., [32–34]. These studies show that primates [34,35], rats [36] and birds [33,37] can successfully achieve synchronization in tasks where they are required to synchronize a motor action such as pressing a lever or pecking a key to an audio or visual metronome stimulus (even with adaptations to changes in tempo). The most flexible rhythm production has been shown by a sea lion and two parrots who were capable of synchronizing to the beat of real music at varying tempos (sea lion: [38,39]; parrots: [40,41]), thus indicating that coupling is not restricted to a certain rate. Even though these lab-based studies offer a very controlled environment to investigate the cognitive flexibility of synchronization abilities, they also heavily rely on motivational factors that do not necessarily relate to how flexible rates of rhythmic abilities can be adjusted. These factors might sometimes be overlooked, and potentially lead to false negative results [3]. The other avenue that has been taken is relying on more naturalistic observation where rhythmic abilities from a species-specific behavioral repertoire are used (e.g., chimpanzee walking: [42]). Critically, this approach can be further enhanced when combined with mathematical modeling [2].

One such context where animals are expected to show high motivation for temporal coordination and that is under high selection pressures, is anti-predator vigilance. There is ample evidence that individuals living in bigger groups spend less time being vigilant [43–45], perhaps because they simply feel safer (i.e., the many eyes effect [46]) or because they actually take others' vigilance into account. For several species it has been shown that individuals' vigilance levels are indeed not independent of each other but rather synchronized at the group level (e.g., birds: [47,48]; mammals: [49–51]), leading to periods of higher and lower vigilance. Intriguingly, vigilance can also be coordinated in anti-phase synchronization (i.e., animals maximizing the feeding time when any group member is vigilant [52]), leading to turn-taking-like patterns [53–55]. Such evidence for anti-phase synchrony is most prevalent in groups with sentinel systems and otherwise typically restricted to (mating) pairs (coral reef fish: [56,57]; birds [58,59]).

For the small, arboreal common marmosets, who are vulnerable to predation in the wild ([60], carnivores: [61], snakes: [62], raptors: [63]), vigilance is a key part of their survival strategy [64]. They follow the general trend of a negative group size effect with bigger groups being associated with lower levels of individual vigilance [65]. Some studies have claimed the presence of a sentinel system [66,67] and more recently, marmoset pairs have shown to coordinate their vigilance in a feeding situation by being more vigilant when the pair mate was feeding than when not [68]. Furthermore, when feeding in proximity to their mate, they maintained the same behavior for longer periods of time when showing opposite behaviors, i.e., one individual being vigilant and the other feeding [68]. These results suggest temporal coordination between individuals, but how this anti-phase synchrony develops or fades out is currently unknown.

Here, we dynamically modeled the vigilance and feeding behaviors of captive pairs of common marmosets as coupled oscillators (i.e., oscillators changing behaviors between vigilance and feeding) using the Kuramoto model. We used a feeding situation where being vigilant and feeding were mutually exclusive (namely, when marmosets were eating mash out of an opaque

feeding bowl). Even in captive settings marmosets maintain high levels of vigilance, as for instance when responding to unfamiliar humans with antipredator behavior and emitting warning calls upon spotting birds of prey (personal observations by RKB & JMB). First, we described the general properties of marmosets' vigilance and feeding bouts, especially comparing mean vigilance and feeding durations between times when animals were situated alone versus together on a feeding basket. Next, we modeled the two individuals as coupled oscillators according to the classic Kuramoto model [69,70] (henceforth 'Kuramoto model'). The Kuramoto model is one way to model the development of synchrony and its temporal variations. Other popularly used models include the integrate-and-fire model for pulse-coupled oscillators (when the oscillators interact only momentarily during a cycle, such as firefly flashes) [71] and the second-order Kuramoto model for oscillators with inertia (such as power grids) [72]. As our system does not fit these special cases, we started modeling our system using the classic Kuramoto model. However, the classic model assumes that the individual oscillators are inherently periodic. We thus additionally developed a non-periodic version of the Kuramoto model, as many biological systems are not expected to be inherently periodic [73,74].

An overview of analyses is summarized in Fig 1 and described in detail in the methods section. Briefly, we started studying marmoset vigilance behavior when they were feeding alone and simulated behavioral bouts based on this data. We then virtually paired the simulated bouts of the marmoset partners and fit the classic Kuramoto model to these instances (Fig 1A). These were the 'control bouts', reflecting patterns of synchronization that would randomly occur. Next, we studied marmoset vigilance behavior when the marmosets were actually feeding together with their partner (and hence, were highly likely to be influenced by their behavior) and fit the classic Kuramoto model to these 'actual bouts' (Fig 1B). Finally, we simulated several marmoset behavioral bouts using the non-periodic Kuramoto model and compared the simulations to empirical data. By fitting the classic and non-periodic Kuramoto models to the data, we were able to estimate how strongly one individual's behavior influenced the other individual, i.e., the coupling strength–through the coupling constant. Positive and negative values of the coupling constant indicate the tendency to reach in-phase and anti-phase synchrony, respectively. We additionally derived the critical coupling constant (the threshold to reach a state of synchrony), using the classic Kuramoto model and the control model. For the classic Kuramoto model, we also obtained the time to reach anti-phase synchrony.

Our predictions were threefold: 1) Most biological oscillators are not perfectly periodic, yet there are abundant examples of synchronization in the animal world. We did not expect marmoset head oscillations (i.e., their behavioral transitions from vigilance to feeding) to be periodic in the absence of any input from conspecifics, but we predicted that they would still show anti-phase synchrony with a partner, given that the probability of an individual being vigilant when its partner is feeding has been shown to be higher than chance [68,73–75]. 2) Animals that show a stereotyped periodic behavior and eventually synchronize, such as fireflies flashing or katydids chirping seem to do so following simple, fixed interaction rules. If marmosets were following simple, fixed interaction rules to coordinate vigilance, we would expect the coupling strengths to be more-or-less uniform across feeding bouts. Moreover, we would expect the synchronization dynamics to be uniform. As marmoset vigilance behavior is known to be variable and not very stereotyped [68], we predicted that there will be some variation in the coupling strengths across feeding bouts and that marmosets can synchronize flexibly. 3) If the coupling strength indeed varied across feeding bouts, we predicted that the differences in the initial rate of head oscillations can explain some of this variation. This was motivated by the fact that marmosets show sensitivity to initial conditions when synchronizing in other modalities. For example, marmosets converge with their pair-mates in acoustic space, i.e., sound

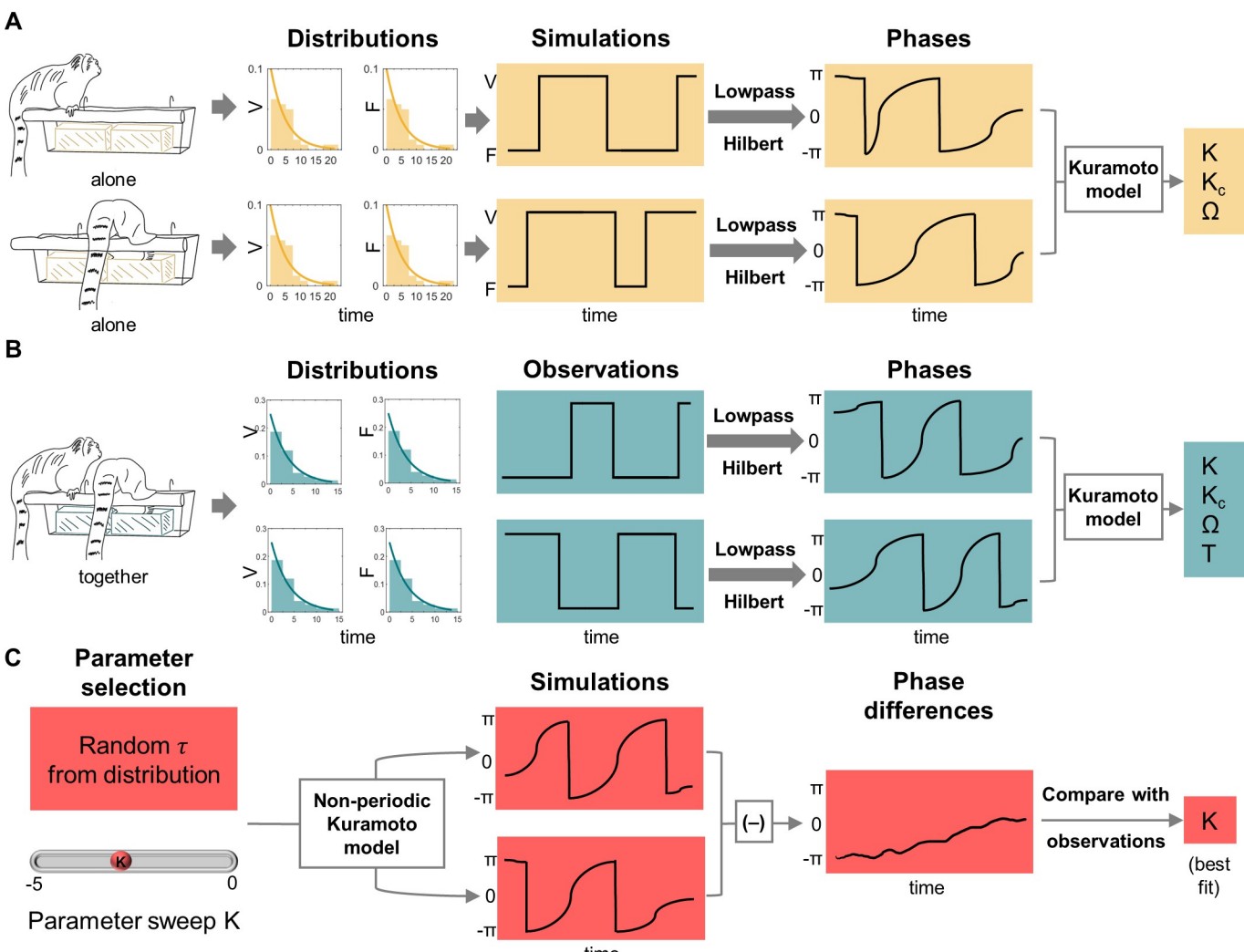

**Fig 1. Overview of analyses. (A) The pipeline for analyzing control bouts**. Individuals were observed when they were feeding alone, and their vigilance and feeding distributions were obtained (an individual sat on the basket and ate from the yellow bowl). The distributions were used to simulate 100 behavioral bouts per pair. These bouts underwent lowpass filtering and Hilbert transform, and the Kuromoto model was fit to the phases to obtain synchronization parameters. **(B) The pipeline for analyzing actual bouts using the classic Kuramoto model**. Individuals were observed when they were feeding together, and their vigilance and feeding distributions were obtained. Additionally, the time-series of when each individual was vigilant or feeding was obtained. These time-series (bouts) underwent lowpass filtering and Hilbert transform and Kuromoto model was applied on the phases to obtain synchronization parameters as in the control condition. **(C) Non-periodic Kuramoto model.** The distributions of vigilance and feeding durations were provided as inputs to the non-periodic Kuramoto model and a parameter sweep of coupling constants done to simulate 2 time-series of phases. From this, the time-series of phase difference was calculated and its angular distance from the time-series obtained from a behavioral bout of the 'together' condition was determined. The coupling constant value, which gave the smallest angular distance (provided the 'best fit') was the estimated coupling constant for that behavioral bout. Note that even though monkeys are depicted to be provided with one food bowl when alone and two food bowls when together, both one bowl and two bowl conditions were experienced by all animals to control for alternative explanations. See the methods sections for further details. Abbreviations: V = vigilance, F = feeding, K = coupling constant, $K_c$ = critical coupling constant, $\Omega$ = difference in natural frequencies, T = time to attain anti-phase synchrony, and $\tau$ = time-period for which an individual would remain in a particular behavioral state.

more similar to their partner over time, in a process called vocal accommodation [76,77]. During this process, they also move through the acoustic space in a synchronized fashion [78]. The extent of vocal accommodation is proportional to the initial differences in vocal properties, with higher initial acoustic differences between partners leading to more vocal accommodation later [79].

## Results

### Properties of marmoset vigilance behavior

The distributions of durations for which an individual displayed a behavior before switching to a different behavior were monotonically decreasing (see Fig 2A for distributions of an example pair and S1 Fig for all individuals). We modelled these switches in behaviors as Poisson

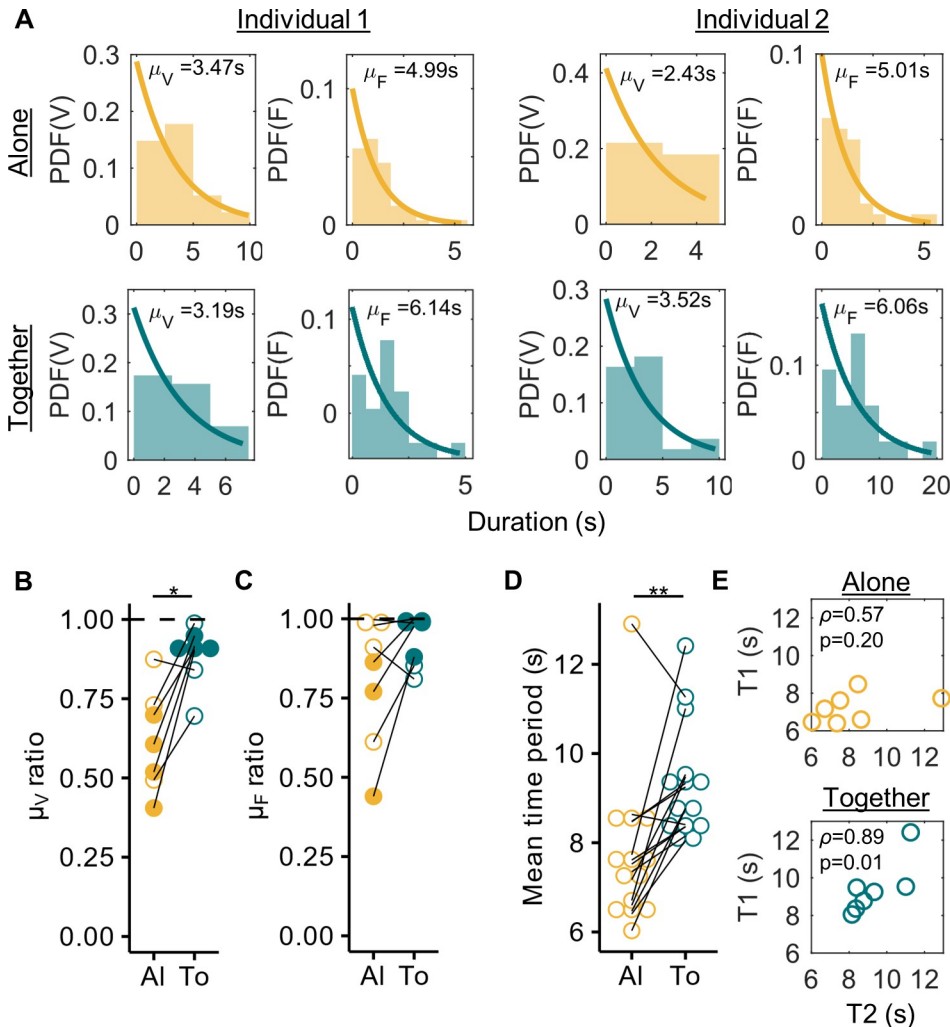

**Fig 2. Properties of marmoset vigilance behavior. (A)** Distribution of behavioral durations for vigilance (V) and feeding (F) behaviors for an example pair in the data. The two left columns correspond to one individual and the two right columns to the other individual of the pair. The top row depicts the distributions when the individuals were alone and the bottom when they were together. Each bin of the histogram is 2.5s. Exponential fits to the probability density functions (PDFs) are shown with their corresponding parameter values (mean $\mu$). **(B, C)** Plots comparing the ratio of mean vigilance durations ($\mu_V$ ratio, **B**) and mean feeding durations ($\mu_F$ ratio, **C**) of the two individuals of the pairs (n = 7 pairs) between the alone ('Al') and together ('To') conditions. Each point is a pair and the same pairs are connected by lines. Values closer to 1 mean that the distributions of durations for the two individuals are similar, a sign of synchrony. Furthermore, if the probability of obtaining the together value by chance from the alone distributions was <0.05 (see Methods), filled circles are shown instead of open circles. *p<0.05, two-sided Wilcoxon signed-rank test. **(D)** Plot compares the mean time period of head oscillations ($\mu_F + \mu_V$) of n = 14 individuals between the alone and together conditions. Each point is an individual, and the same individuals are connected with lines. **p<0.01, two-sided Wilcoxon signed-rank test. **(E)** Correlation between the mean time periods ($\mu_F + \mu_V$) of the individuals of the pair in alone (top) and together (bottom) conditions. Spearman's $\rho$ and the corresponding p-values are shown.

processes and therefore fit exponential functions to the distributions of durations. We found the fit parameter μ for vigilance distributions to be more similar between individuals of a pair (ratio between the individuals significantly higher and closer to 1) when they were together compared to when they were alone (Fig 2B, n = 7 pairs, signed-rank = 1, p = 0.03, two-sided Wilcoxon signed-rank test). Values closer to 1 suggest synchrony. Moreover, for four pairs, the probability of obtaining a ratio as extreme as in the 'together' condition from the 'alone' distributions by chance was <0.05. Significant differences were not found for feeding distributions (Fig 2C, n = 7 pairs, signed-rank = 4, p = 0.11, two-sided Wilcoxon signed-rank test). However, even in this case, the probability of obtaining a ratio as extreme as in the 'together' condition from the 'alone' distributions by chance was <0.05 for three pairs.

While ratios of fit parameters alluded towards synchrony in the 'together' condition, they cannot indicate the nature of synchrony, i.e., whether the marmosets showed in-phase or anti-phase synchrony. For this, we compared the estimated mean time-period of head oscillations ($\mu_F + \mu_V$) between the two conditions and found it to be significantly higher when the individuals were together (Fig 2D, n = 14 individuals, signed-rank = 8, p = 0.0031, two-sided Wilcoxon signed-rank test). This effect was mostly driven by a significant increase in the feeding durations of the individuals (S2 Fig). It indicated that the individuals were most likely in anti-phase synchrony when together. More evidence for synchrony came from the fact that the mean time period of head oscillations between the individuals of a pair was positively correlated when they were together (Fig 2E bottom panel, n = 7 pairs, Spearman's coefficient = 0.89, p = 0.01) and not when they were alone (Fig 2E top panel, n = 7 pairs, Spearman's coefficient = 0.57, p = 0.2).

## Formulating the models

The Kuramoto model assumes that the individual oscillators are periodic. The Kuramoto model for N oscillators is given by:

$$\frac{d\theta_i}{dt} = \omega_i + \frac{K}{N} \sum_{j=1}^{N} \sin\left(\theta_j - \theta_i\right), i = 1 \ldots N \tag{1}$$

Where $\theta_i$ is the phase of the i[th] oscillator, $\omega_i$ is the natural frequency of the i[th] oscillator, K is the coupling constant and t is time.

From Eq (1), we propose an extension to the Kuramoto model for non-periodic oscillators. This extension is similar to the extension to the integrate-and-fire model for non-periodic fireflies proposed by Sarfati et al. [74]. Let there be N non-periodic oscillators, each going through a sequence of M states to complete one oscillation and the state of the i[th] oscillator at time t be represented by $\Psi_i$. Suppose the probability of transitioning from one state to all other states except the next one in the sequence is 0. Let the duration of the i[th] oscillator to remain in the p[th] state before transitioning to the next one be denoted by $T_{p,i}$, be independent of its duration to remain in any other state, and follow the distribution $g(T_{p,i})$. Then Eq (1) can be modified to:

$$\frac{d\theta_i}{dt} = \frac{2\pi}{\sum_{p=1}^{M} \tau_{p,i}} + \frac{K}{N} \sum_{j=1}^{N} \sin\left(\theta_j - \theta_i\right), i = 1 \ldots N$$

$$\tau_{p,i}(t) = \begin{cases} \tau_{p,i}(t-1), \Psi_i(t) = \Psi_i(t-1) \\ X_{p,i,t} \sim g\left(T_{p,i}\right), otherwise \ (\ including\ when\ t = 0) \end{cases} \tag{2}$$

where $X_{p,i,t}$ is a random number drawn from the distribution $g(T_{p,i})$ at time t and $\tau_{p,i}$ is a random variable who's value changes whenever the state of the oscillator changes.

Simulations of Eq (2) for the case of marmosets where two non-periodic oscillators switch between two states (see Methods) show that non-periodic oscillators can display both in-phase and anti-phase synchrony depending on the value of the coupling constant (Fig 3).

## Results from model fits

Out of the 53 behavioral bouts from 7 marmoset pairs in the data, the Kuramoto model could only be fit to 46 bouts, the remaining 7 bouts being either too short in duration (<10s) or consisting of too few head oscillations (<2 for both individuals) (see S3 Fig for the histogram of bout durations before and after discarding the 7 bouts). Along with those seven bouts, we also discarded all other bouts that were less than 10s long or in which both individuals underwent less than 2 complete head oscillations. This provided us with 38 bouts to analyze.

The coupling constant (K) from the Kuramoto model fit was more negative than the critical coupling constant ($K_c$) for 35 out of the 38 bouts (i.e., 92.1% of all bouts, Fig 4A and 4B) which thus ended in anti-phase synchrony. Based on this model, the average time required for each pair to reach anti-phase synchrony was 6.84±2.27s (mean±sd). Overall, the fraction of bouts

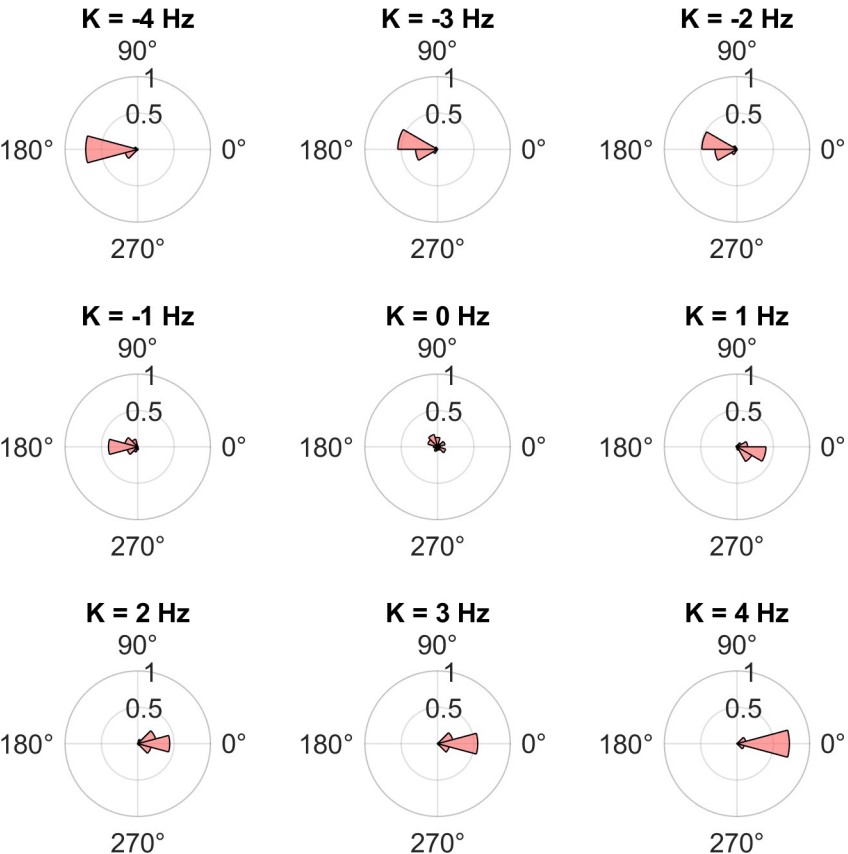

**Fig 3. In- and anti-phase synchrony in the non-periodic Kuramoto model.** Example polar probability histograms of the phase differences between 2 non-periodic Kuramoto oscillators (see Eq 11) for different values of the coupling constant (K). A phase difference of 0 corresponds to in-phase synchrony, and that of 180˚ ($\pi$ radians) corresponds to anti-phase synchrony. The parameter values for the model simulations were the average of values in the data ($\mu_{F,1}$ = 5.06s, $\mu_{V,1}$ = 3.17s, $\mu_{F,2}$ = 4.04s, $\mu_{V,2}$ = 3.10s, duration of simulation = 53.32s = 1333 time steps), and the initial phase difference was $\pi$/2.

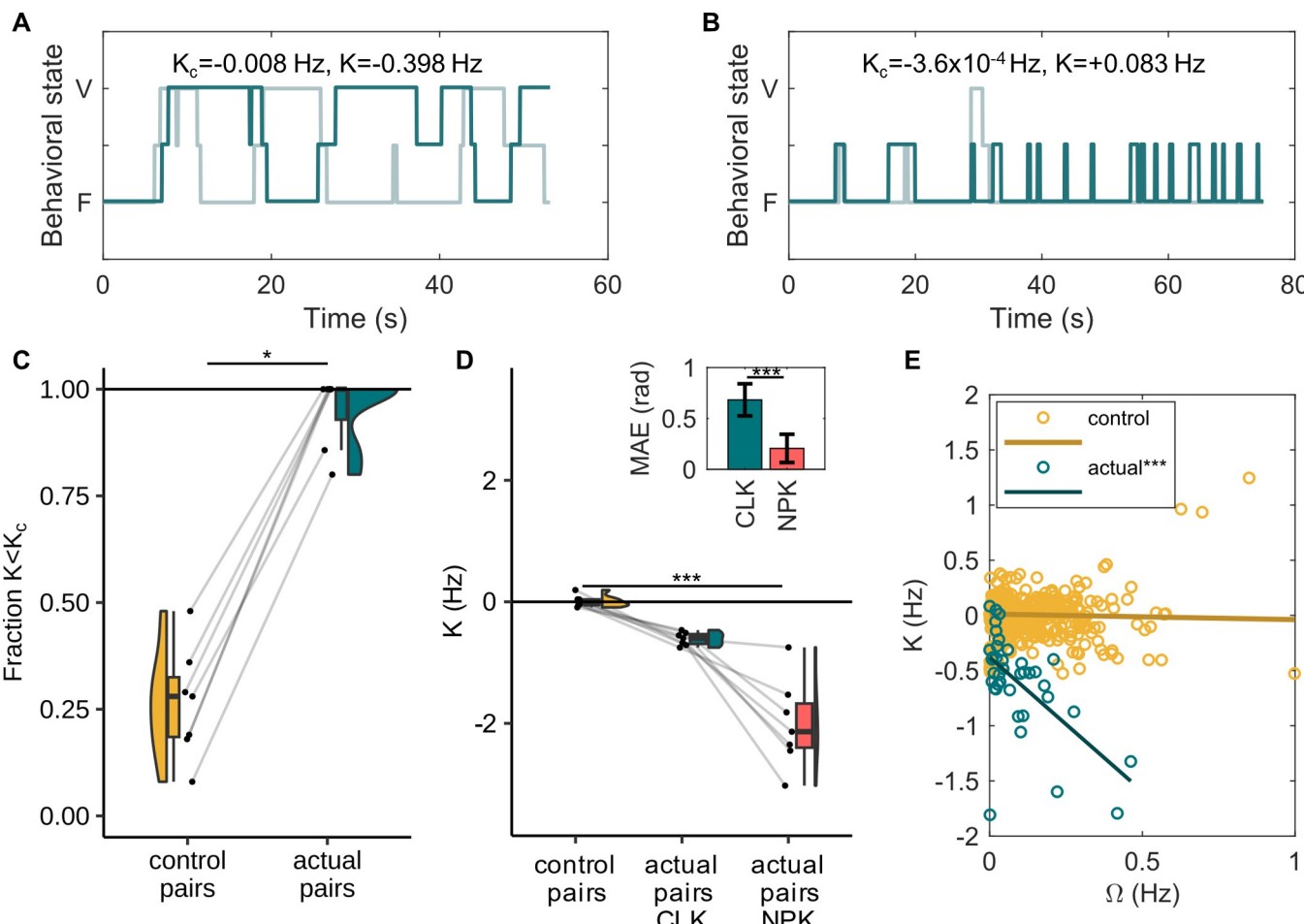

**Fig 4. Synchrony in control and actual pairs. (A, B)** Example behavioral bouts from actual pairs that do (A) and do not (B) take turns. Plots show time-series of behaviors of the two individuals of a pair when they were together. Each curve (dark or light teal) corresponds to one individual. 'V' stands for vigilant state and 'F' for feeding. The state in between refers to all the instances when the animal's line of sight could not be classified by the experimenter as either vigilant or feeding and was coded as 'out of sight' (see S2 Table for the definition). Here, $K_c$ and $K$ are obtained from the classic Kuramoto model fit. 92.1% of the analyzed data (from actual pairs) consisted of bouts such as in (A) where $K<K_c$ and only 7.9% consisted of bouts such as in (B) where $K>K_c$. **(C)** Fraction of behavioral bouts out of total for which the coupling constant was less (more negative) than the critical coupling constant for control and actual pairs (n = 7). *p<0.05, two-sided Wilcoxon signed-rank test. **(D)** Raincloud plots depict the coupling constant (K) in control pairs, actual pairs with the classic Kuramoto model (CLK) fit and actual pairs with non-periodic Kuramoto model (NPK) fit. For each group, every point shown is the mean of K-values of all bouts of that pair (n = 7 pairs), the box plots summarize the statistics, and the half violin plots show the distribution of the data. ***p<0.001, post-hoc Nemenyi test. The bar plot on the top right shows the average mean absolute error (MAE) ± s.d. values in radians for CLK and NPK model fits. ***p<0.001, Wilcoxon signed-rank test. **(E)** Relationship between the critical coupling constant and the difference in natural frequencies of the two individuals for control bouts (simulated, n = 700) and actual bouts (n = 38). Lines show linear regression fits to control and actual bouts. ***p<0.001, non-parametric ANCOVA.

for which K<Kc was significantly higher for actual pairs than control pairs (Fig 4C, signed-rank = 28, p = 0.016, two-sided Wilcoxon signed-rank test). This suggests that given enough time, in all these instances when K<Kc, the marmoset pair would evolve into an anti-phase pattern of vigilance. The K-values of the control pairs, actual pairs with the classic Kuramoto model fit, and actual pairs with the non-periodic Kuramoto model fit were significantly different (Fig 4D, $X^2$ = 14, df = 2, p<0.001, Friedman test). Post-hoc analysis showed that control pairs and actual pairs with the non-periodic Kuramoto model fit differed significantly (control pairs–non-periodic Kuramoto: p<0.001, control pairs–actual pairs (Kuramoto): p = 0.147, actual pairs (Kuramoto)–non-periodic Kuramoto: p = 0.147, post-hoc Nemenyi test). As expected, the non-periodic Kuramoto model provided a superior fit compared to the classic

model with significantly lower Mean Absolute Error (MAE) values (Fig 4D, n = 39 behavioral bouts, signed rank = 739, p<0.001, Wilcoxon signed-rank test). Further, there was no effect of the location or the number of bowls in themselves, but we did find a significant effect of their interaction on the K-value (S1 Table). This was driven by a more negative K-value when the marmosets were outside and provided with two bowls (S4 Fig). The Gaussian linear mixed-effect model was significantly different from the null model (log-ratio = 11.18, p = 0.01).

Across the 38 bouts, the K-value from the classic Kuramoto model negatively covaried with the difference in natural frequencies (Ω) between the individuals of the pair (Fig 4E, slope of linear regression = -2.43, y-intercept = -0.38 Hz, RMSE = 0.36 Hz). Linear regression analysis of K-values of control bouts (n = 700) and how they varied with Ω gave negligible slope and intercept (Fig 4E, slope = -0.05, y-intercept = 0.01 Hz, RMSE = 0.49 Hz). Non-parametric analysis of covariance (non-parametric ANCOVA) showed that the two regression lines were unequal (error standard deviation = 0.22, p<0.001). Further, there was no effect of the location (inside/outside) or the number of bowls (one/two) on the regression lines (non-parametric ANCOVA, error standard deviation = 0.32, p = 0.233).

## Discussion

In this study, we show that pairs of marmosets coordinate vigilance by displaying anti-phase synchrony in their behaviors despite each individual being non-periodic. More interestingly, we find that coupling between individuals is flexible and there is increased coupling when the individuals are inherently more asynchronous. Below we discuss its implications for marmoset behavioral ecology and a plausible mechanism for the above phenomena, and the wider applicability of our mathematical framework for the study of synchronization in nature.

### Implications for marmoset behavioral ecology

Marmosets are not perfectly periodic oscillators when it comes to displaying vigilance and feeding behaviors. When alone, the distributions of durations for which each individual marmoset was vigilant or feeding before changing its behavior followed an exponential distribution (see the alone condition in Fig 2A for an example and S1 Fig for the overall distributions). This property made it ideal to model the behavior as a Poisson process. It suggests that marmosets have an underlying rate of change of behavior (vigilance/feeding); however, each transition in behavior is more or less independent of the other.

Marmosets displayed changes in behaviors when they were together. The distributions of durations of being vigilant between the two individuals of a pair became more similar when together compared to when alone (Fig 2B), a sign of synchrony. The feeding distributions also became more similar in the 'together' condition, even though this effect was not significant (Fig 2C). Overall, the mean time period of oscillations (vigilance and feeding) of the two individuals of a pair were positively correlated when together, which was not the case when alone (Fig 2E). This suggests that becoming similar to each other in vigilance durations along with a moderate level of similarity in feeding durations was sufficient to display overall synchrony. Moreover, marmosets spent longer in a behavioral state before changing their behavior when together, effectively slowing down head oscillations compared to when alone (Fig 2D). Such slowing down of oscillations has been proposed to be a phase-delay mechanism of attaining anti-phase synchrony and has been described in case of alternating singing in several insect and frog species [1,19]. In katydid pairs for example, it has been proposed that a chirp from one individual delays the chirping of the other individual by a tiny bit (less than one cycle), thus inducing a phase-delay [80]. This applies symmetrically to both individuals and over time the individuals display anti-phase synchrony, with the frequency of chirping of each individual

reduced. Based on the slowing down of head oscillations, followed by the confirmation of negative coupling from the Kuramoto model fit to the data (Fig 4D), a mechanistic explanation could be that when a marmoset sees another individual change its behavioral state (feeding/vigilant), it remains in its current state for slightly longer than usual. This applies symmetrically to both individuals and the accumulation of phase-delays ultimately leads to anti-phase synchrony. As to why marmosets may remain in a state for a prolonged duration when together, an ethological explanation could be that a marmoset may feel 'safer' in the presence of the other individual, hence prolonging the time it would feed [56,59,81,82]. The other individual, if vigilant, would now have to accommodate to this by prolonging its vigilance state, altogether triggering a chain of phase-delays that would ultimately lead to anti-phase synchrony. This ethological hypothesis is supported by the fact that the increase in feeding durations was the primary contributor to the slowing down of head oscillations (S2 Fig), whereas accommodating to each other's vigilance durations was the primary contributor to developing and maintaining synchrony (Fig 2B and 2C).

A negative coupling alone will not lead to anti-phase synchrony. The coupling must be strong enough to overcome the mutual differences in natural frequencies of the two individuals that prevent them from synchronizing. By fitting the Kuramoto model, we found that in 92.1% of the analyzed bouts, the coupling between individuals was strong enough to overcome the critical threshold to attain anti-phase synchrony, almost four-fold higher than what was expected by chance (Fig 4C), hence confirming prediction 1 (see Introduction). It means in 92.1% of the observed cases, marmosets would evolve into anti-phase synchrony. Moreover, the coupling constant estimates from the non-periodic Kuramoto model fit were more negative than the Kuramoto model fit. This is intuitive, as the marmosets would have to couple more strongly to display the same temporal progression of synchrony if they were not inherently periodic. Furthermore, the coupling was stronger when the marmosets were outside and had access to two bowls (S4 Fig). The stronger coupling when outside can be attributed to the increase in perceived risk that could have motivated them to attain anti-phase synchrony sooner. The experimental contrast between 1 vs 2 feeding bowls was included to control for the possibility that the animal would be forced to take turns because of a lack of space to feed together. This was clearly not the case, because turn-taking was also present, and indeed stronger, when feeding space was not restricted in the two bowl condition.

In line with prediction 2 (see Introduction), marmosets displayed a high degree of flexibility in coupling. The coupling strength ranged from -1.81 Hz to +0.08 Hz. This variability was not random; the more the difference in natural frequencies was between two pair mates, the stronger and more negative was the coupling strength (Fig 4E). Needless to say, this is not expected to happen by chance (see control bouts in Fig 4E) as differences in natural frequencies and coupling strengths are two independent parameters of coupled oscillator systems. Nor does the simple mechanistic description (see above) explain such a trend. Instead, marmosets seem to put more effort in synchronizing with each other (by coupling more strongly) when they begin in a more desynchronized state, confirming prediction 3 (see Introduction). This makes the process cognitively more demanding as it would require the marmosets to estimate the amount of asynchrony i.e., the differences in rhythms, and mutually compensate for larger differences in rhythms with a stronger coupling. However, currently, we do not have enough understanding of the cognitive aspect of synchronization in marmosets to claim if this is a conscious process (e.g. the represented goal to achieve anti-phase synchrony) or otherwise.

Intriguingly, similar patterns are found when marmosets accommodate to their partners in other modalities as well. For example, marmosets have been shown to converge to their partner in vocal space over time [77,79] (called "vocal accommodation" [76]) and it has been observed that the more the individuals differ in vocal properties initially, the more they

converge. The simplest mechanistic explanation obtained from mathematical modeling of the phenomenon suggests the presence of a dynamic vocal learning mechanism [78], which is again cognitively more demanding than simple forms of learning. Overall, it is clear that mathematical modeling is a powerful tool that could help us tap into the flexibility and control that animals might have over synchrony parameters and, ultimately, the cognitive aspects of synchrony. Marmosets with their ability to flexibly synchronize with other individuals in various modalities such as in motor coordination [83] (for evidence of motor coordination in another other callitrichid species see [84]), vocal turn-taking [12,13,85,86] and peripheral oxytocin levels [87] make for an ideal animal model to study this phenomenon. [83,84].

We acknowledge that this study has been conducted in captivity where vigilance is present [65,67,88–90] but chronic threat levels are much lower than in the wild. We hypothesize that under higher predation pressure, coordinated vigilance is even more likely to emerge since animals would experience much stronger trade-offs between the need to be vigilant and other activities. However, teasing apart the turn-taking coupling pattern in naturalistic conditions would not have been possible because wild marmosets hardly ever feed head-down without seeing anything at all (they feed on a wide-ranging diet, including insects, fruits, small mammals, but, importantly, also exudates [91]). This contrasts with the experimentally induced setting in this study, where the feeding individuals' views were entirely obstructed which allowed us to fully separate vigilance and feeding behavior. It would be highly desirable to complement the findings from our study with data from wild populations of common marmosets. Moreover, it is likely that vigilance coordination, both in captivity and in the wild, also occurs in other contexts, such as when infants are playing [92,93].

## Implications for studying synchrony in nature

Weakly coupled oscillators are prominent motifs in the animal world, exemplified by gatherings of fireflies flashing together [4], crickets producing synchronized chirps [19], and animals calling antiphonally [12,14,15]. These systems take a finite time to attain synchrony if they begin in an unsynchronized state. Therefore, if an observer observes the system before it attains synchrony, they will not see the system to be synchronized but can only note that the system is tending towards synchrony. In such cases, looking at mean phase coherence, distribution of phase differences, or overall correlation may not necessarily show this trend. By fitting the Kuramoto model to the system, one can detect such a trend towards synchrony. Moreover, for the n = 2 oscillators case, we show that it is possible to estimate the time required for the system to attain synchrony (see Eq 11 in methods), which allows for assessing the biological relevance for the system under study. For example, if it would take 30 minutes instead of 6 seconds for the marmosets to attain anti-phase synchrony, this could hardly be a good predator avoidance strategy during feeding bouts, even though the coupling constant from the Kuramoto model fit in such a case would be more negative than the critical coupling constant.

A drawback of the Kuramoto model is that it assumes that each oscillator it is modelling is inherently periodic. However, in biology, we seldom come across perfectly periodic oscillators. The Kuramoto model can indeed tolerate some level of variability in natural frequencies, but when the distributions are not narrow, as in the case of *Photinus carolinus* fireflies' inter-flash intervals [94] or vigilance and feeding durations of marmosets, modifications to previous models are required. Akin to Sarfati et al.'s extension to the integrate-and-fire model [74], our extension to the Kuramoto model allows users to implement it even when individual oscillators are not perfectly periodic and makes the model applicable to a much wider spectrum of biological phenomena. While the non-periodic model fit provides a more realistic estimate of

the coupling strength, estimating other parameters, such as the critical coupling constant or time required to attain synchrony, is not straightforward. Additionally, unlike the classic Kuramoto model, one cannot 'fit' the non-periodic Kuramoto to the data in the traditional sense. Estimating the coupling strength requires simulating the model several times by performing a parameter sweep. We, therefore, suggest using the classic and non-periodic models as complementary approaches, as, along with providing critical coupling constant and time to attain synchrony estimates, the classic Kuramoto model also helps narrowing down the range of coupling constant values to perform the parameter sweep. In closing, we emphasize the utility of mathematical modeling and its power of abstraction to better understand complex biological processes. We hope that by adding to the long list of demonstrations of the capabilities of mathematical abstraction, we inspire more and more biologists to harness this tool to advance and enrich their field.

## Methods

### Ethics statement

All experiments were carried out in accordance with the Swiss legislation and were approved by the Kantonales Veterinäramt Zürich under the license number ZH223/16 with the degree of severity 0.

### Subjects and housing

The behavioral data for this study was collected by Brügger et al. (2023) [68] on five breeding pairs, a pair of adult siblings and a family group with young infants (age of immatures during data collection 1–4 weeks), resulting in a total of 14 adult marmosets (for details on the group composition and testing order see S3 Table). The animals were housed in heated indoor enclosures with access to outdoor enclosures (during appropriate weather conditions: temperature above 10˚ C). All enclosures were equipped with climbing materials (natural branches, ropes etc.), resting platforms, hammocks, sleeping boxes, and a bark mulch-covered floor. Feeding of the animals occurred at least twice a day in the mornings (ca. 8:00) with a vitamin enriched mash and during lunch (between 11:00–12:00) with fresh vegetables. In the afternoons, animals received a snack feeding with various additional protein sources (eggs, nuts, insects). Marmosets had access to water ad libitum.

### Procedure

Data was recorded during the regular morning feedings without an experimenter being present to eliminate the effects of vigilance towards the experimenter. Therefore, animals were filmed with one or two video cameras (Sony HDR-CX730/HDR-CX200) placed inside or outside of the home enclosures. The cameras ran for about 15–30 minutes to ensure full feeding sessions were captured (as not all animals were feeding at the same speed).

To control for environmental factors that could potentially influence vigilance levels, we implemented a two by two design. We varied 1) the location of the feeding between 'inside' (i.e., the inside part of the home enclosure, where animals are normally fed) and 'outside' (i.e., the outside part of the home enclosure, accessible via a semi-transparent tube system) and 2) the number of feeding bowls the mash was provided in between one or two bowls while keeping the amount of food constant. The location of the feeding was altered to provide variation in the experienced risk level of the feeding situation. 'Inside', where animals were used to being fed, leading to a "lower risk" situation and 'outside', where potential predators (such as cats, birds of prey or more rarely foxes) were visible more frequently, leading to a "higher risk"

situation. We provided one or two feeding bowls, this was necessary to account for the potential alternative explanation that turn-taking was externally imposed by space restriction, i.e. that the marmosets avoided to have the head inside the same feeding bowl simultaneously with their partner and therefore opportunistically engaged in vigilance while waiting. Critically, if the same turn taking pattern also emerged in the two bowl condition, this alternative explanation can be excluded [68]. Moreover, the food (mash) was only accessible via licking from the bottom feeding bowls. The rims of the bowls were higher than the head of the marmosets thus leading to full coverage of the eyes of the animals while feeding, making feeding and vigilance behavior mutually exclusive. For an overview of the experimental setup see S5 Fig. The animals experienced the four conditions in a semi-randomized order, for a total of 28 sessions collected over 5 weeks in May 2018.

## Data coding

Video coding was done with the software INTERACT (Mangold GmbH, version 18.0.1.10, Arnstorf, Germany). Coding started with the first frame from which the experimenter was no longer visible (location 'inside') or the frame when the first individual was fully (all four limbs) located outside of the tube that connected the inside to the outside enclosure (location 'outside'). The following time periods were excluded from coding: 1) when animals were not feeding for more than 4 min. 2) when obvious outside disturbances occurred that would externally induce vigilance, i.e., cats walking past the outside enclosure, other groups of the colony vocalizing loudly (as we were interested in the effects of pre-emptive vigilance, not reactionary vigilance, see [95]). Sessions were deemed finished when animals stopped eating because all the mash had been eaten, or when they interrupted feeding for more than 4 min. without resuming, or a maximum of 10 min after the start of the session. We coded three behaviors frame by frame (videos were recorded with 25 fps) namely: vigilance, feeding and out of sight as well as the locations of the animals (inside/outside or on basket) according to the detailed definitions in S2 Table. Interobserver reliability was assessed by coding 21% of all video data by a second rater. We reached interclass correlation coefficients (ICC3) of 0.95 (95% confidence interval, CI [0.86, 0.98]) for vigilance, 1 (95% CI [0.99, 1.00]) for feeding and 0.94 (95% CI [0.85, 0.98]) for the location categories.

## Data preparation for analyses

We used the onsets and offsets of behaviors "vigilance" and "feeding" coded from the videos (according to definitions in S2 Table) to determine behavioral bouts of interest in all four conditions (inside 1 bowl, inside 2 bowls, outside 1 bowl, outside 2 bowls). Analysis was restricted to behavioral states when (1) *both* animals were located on the feeding basket ('together'; to fit the Kuramoto Model and the non-periodic Kuramoto model, see below) or (2) one animal was located *alone* on the feeding basket ('alone'; for control simulations). Since animals were allowed to move freely in the enclosures, time spent on the basket occurred in bouts (see also S5 Fig). We determined bout length by studying the distribution of breaks in behavior (i.e., whenever an animal was not displaying vigilance or feeding) and assigned 10s to be a sensible cutoff. These criteria resulted in a total of 53 behavioral bouts from 7 pairs when both individuals were together on the feeding basket (7.6±4.0 mean±s.d bouts per pair). The behavioral bouts were converted to a numerical time series by assigning the following values to the behavioral states: vigilance = 1, feeding = 0. Marmosets also showed behaviors that could not be assigned either to the definitions of vigilance or feeding. These behaviors were considered transition states between vigilance and feeding and were assigned the value of 0.5 (this included behaviors where the line of sight of the individual was not clear, i.e., "out of sight").

                                      

Because marmosets are known to show head movements of angular velocities up to 1000 degrees per second [96], which equates to 2.78 complete head oscillations per second, we low-pass filtered the time series using a filter of 2.78 Hz (using MATLAB's 'lowpass' function from the 'signal processing toolbox'). This smoothened the times series by modeling the transitions between the feeding and vigilance states to be gradual. A Hilbert transform was applied to the filtered time series (using MATLAB's 'hilbert' function from the 'signal processing toolbox'), and a time series of phase angles was obtained. This helped map marmoset head oscillations onto a circle wherein instantaneous head positions could be represented by unique phase angles ranging from -π to π radians.

## Investigating the properties of marmoset vigilance behavior

We investigated the durations for which an individual showed a particular behavior before switching to a different behavior. Previously, researchers have fit various probability distributions to infer the patterns underlying a species' vigilance behavior. The Normal, Log-Normal and Exponential distributions turn out to be the best fitting distributions under different cases [97]. In species that show a regular, periodic vigilance, characterized by a distinct peak in the distribution for a particular duration of vigilance, the Normal distribution turns out to be the best fitting function. In some species, the vigilance distributions are right-skewed and the log-normal distribution provides a superior fit. Researchers have hypothesized that this in would indicate that various factors underlying the vigilance behavior of the species have a multiplicative effect. In a few cases, the vigilance distributions are monotonically decreasing, and the negative exponential distribution turns out to be the best fit. This would suggest a Poisson-like process with an underlying rate of switching between behaviors such that each switch is independent of previous switches. The distribution of vigilance durations for marmosets followed a monotonically decreasing function, and we therefore fit exponential functions (using MATLAB's 'fitdist' function from the 'statistics and machine learning toolbox') to the distributions of durations. The toolbox fits the function:

$$y = \frac{1}{\mu} e^{-\frac{x}{\mu}}$$

where the fit parameter μ is simply the mean duration. We studied the fit parameter for the two behavioral durations for all individuals, both when they were alone and when they were together. Specifically, we compared the ratio of the fit parameters of the two individuals of a pair for vigilant and feeding durations. A ratio closer to 1 would indicate that the two individuals have similar distributions, whereas a ratio away from 1 would indicate that the two individuals have dissimilar distributions. We also drew 1000 bootstrap samples from the vigilant and feeding distributions when individuals were 'alone' and determined the distribution of the fit parameter ratios (for both vigilance and feeding). We then determined the Z-score and the corresponding p-value of the fit parameter ratio obtained from the 'together' condition based on its distribution in the 'alone' condition. This represented the probability of obtaining a ratio as extreme as the one from the 'together' condition from the 'alone' condition by chance. Then, for each individual, we estimated the mean time period of head oscillations to be the sum of the vigilance and feeding fit parameters and compared this time period between the 'alone' and 'together' conditions. Finally, we checked if the mean time-period of head oscillations was correlated between the individuals of the pair in both the 'alone' and 'together' conditions.

## Fitting the Kuramoto model

The classic Kuramoto model [69,70] for N oscillators is represented by the equation:

$$\frac{d\theta_i}{dt} = \omega_i + \frac{K}{N}\sum_{j=1}^{N}\sin\left(\theta_j - \theta_i\right), i = 1 \ldots N \tag{1}$$

Wherein the i$^{th}$ oscillator completes its cycles at a rate of its natural frequency $\omega_i$ (in cycles per second = Hertz) and its cycles can be mapped onto a circle and its instantaneous state represented by the phase $\theta_i$.

K is the coupling constant and t is time. For the case of two oscillators, it boils down to the following system of equations:

$$\frac{d\theta_1}{dt} = \omega_1 + \frac{K}{2}\sin(\theta_2 - \theta_1) \tag{3}$$

$$\frac{d\theta_2}{dt} = \omega_2 + \frac{K}{2}\sin(\theta_1 - \theta_2) \tag{4}$$

Suppose $\omega_2 > \omega_1$. Subtracting Eqs (3) from (4), we get:

$$\frac{d\theta_2}{dt} - \frac{d\theta_1}{dt} = \omega_2 - \omega_1 - K\sin(\theta_2 - \theta_1) \tag{5}$$

We substitute $\theta_2$ - $\theta_1$ with $\Phi$ and $\omega_2 - \omega_1$ with $\Omega$ to get:

$$\frac{d\Phi}{dt} = \Omega - K\sin(\Phi) \tag{6}$$

We fit Eq (6) to behavioral bouts when both the individuals in a pair were visible by solving it using Euler's method:

$$\Phi(t) - \Phi(t-1) = \Omega - K\sin(\Phi(t-1)) \tag{7}$$

In our case, because the frame rate of the video was 25 fps, the time step size was 0.04s. From the behavioral bouts, we calculated the change in phase difference between the 2 individuals with every frame ($\Phi(t)—\Phi(t-1)$) and divided it by 0.04 to obtain the value in Hertz. We then regressed $\Phi(t)—\Phi(t-1)$ in Hertz on $\sin(\Phi(t-1))$ and obtained the slope and the intercept (using MATLAB's 'fitlm' function). *K* was equal to the negative of the slope and $\Omega$ to the intercept. A positive *K* indicates the tendency to attain synchrony in-phase and a negative *K* to attain anti-phase synchrony, given the magnitude of *K* is greater than a threshold—the critical coupling constant $K_c$.

## Determining the critical coupling constant

To determine the critical coupling constant, we visualized the bifurcation diagram (see chapter 3 in [98]) for Eq (6) against the control parameter $K/\Omega$ within the limits [–5, 5]. Setting $d\Phi/dt = 0$ gives:

$$\Phi = \sin^{-1}\left(\frac{1}{K/\Omega}\right) \tag{8}$$

From the bifurcation diagram (Fig 5), we see that saddle-node bifurcations occur at $K/\Omega$ = -1 and $K/\Omega$ = 1. Stable equilibrium states (depicted by solid black curves) exist under conditions

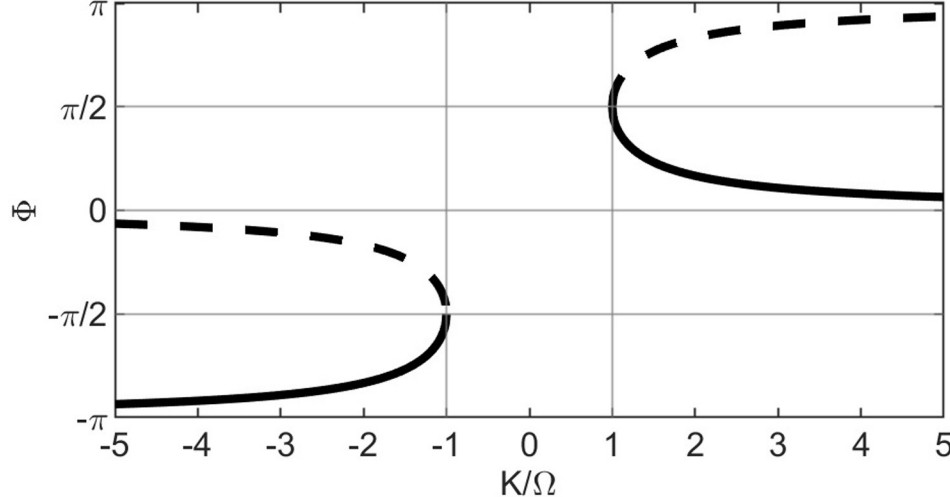

**Fig 5. Bifurcation diagram of the Kuramoto model for 2 oscillators.** The diagram depicts equilibrium points, i.e., the values of ($K/\Omega$, $\Phi$) for which $d\Phi/dt = 0$. Equilibrium points form continuous curves and are of 2 types: stable, represented by solid curves and unstable, represented by dashed curves. The sign of $d\Phi/dt$ in the close vicinity of the solid curves will cause the system to move towards the curve whereas, in the vicinity of dashed curves, will cause the system to move away from it. Note that the y-axis can be wrapped around a circle as $\pi = -\pi$ on a polar plot, and the same graph can be plotted on the surface of a cylinder.

$K/\Omega < $ -1 or $K/\Omega > 1$. The former condition would cause the system to asymptotically approach anti-phase-like synchrony (as $K/\Omega \to -\infty$, $\Phi_{stable} \to -\pi$) and the latter in-phase-like synchrony (as $K/\Omega \to \infty$, $\Phi_{stable} \to 0$). As we were interested in anti-phase synchrony, the critical coupling constant $K_c$ would have to follow:

$$\frac{K_c}{\Omega} = -1 \; or,$$
$$K_c = -\Omega \tag{9}$$

This works, given our initial assumption that $\omega_2 > \omega_1$. However, in general,

$$K_c = -|\Omega| \tag{10}$$

We determined the value of $K_c$ for every bout and compared it with the value of K obtained for that bout. Anti-phase synchrony is attained when $K < K_c$.

### Estimating the time to achieve anti-phase synchrony

To estimate the time taken by the marmoset pair to achieve anti-phase synchrony, we first determined the initial phase difference $\Phi_i$ between the two individuals from the time series of phase angles (see paragraph on data preparation for analyses). We assumed that at time t = 0, the phase difference between the individuals $\Phi = \Phi_i$ and at t = T, they reach anti-phase synchrony, i.e., $d\Phi/dt = 0$ or, from Eq (8), $\Phi = \sin^{-1}(\Omega/K)$. To calculate T, we integrate Eq (6) as

follows:

$$\int_0^T dt = \int_{\Phi_i}^{\sin^{-1}(\Omega/K)} \frac{1}{\Omega - K \sin \Phi} d\Phi$$

$$t \Big|_0^T = \frac{2}{\sqrt{\Omega^2 - K^2}} \tan^{-1}\left(\frac{\Omega \tan\left(\frac{\Phi}{2}\right) - K}{\sqrt{\Omega^2 - K^2}}\right) \Bigg|_{\Phi_i}^{\sin^{-1}(\Omega/K)}$$

$$T = \frac{2}{\sqrt{\Omega^2 - K^2}} \left[ \tan^{-1}\left(\frac{\Omega \tan\left(\frac{\sin^{-1}(\Omega/K)}{2}\right) - K}{\sqrt{\Omega^2 - K^2}}\right) - \tan^{-1}\left(\frac{\Omega \tan\left(\frac{\Phi_i}{2}\right) - K}{\sqrt{\Omega^2 - K^2}}\right) \right] \tag{11}$$

From the values of $\Phi_i$, $\Omega$ and $K$ obtained for every bout, we calculated the time taken to reach anti-phase synchrony using Eq (11).

## Control simulations

We simulated 100 behavioral bouts for every marmoset pair based on the behavior of each of the individuals of the pair when they were situated alone on the feeding basket. The duration of every simulated bout was set to a random number obtained from an exponential distribution fit to the histogram of bout durations of the pair of interest. Then, for each individual in the pair, we determined the distribution of durations between a feeding-vigilant transition and the subsequent vigilant-feeding transition ('vigilant durations') and, similarly, obtained the 'feeding durations'. Next, we simulated behavioral bouts for each individual of the pair separately, considering only its vigilant and feeding durations, and no coupling between them. For each pair, we initiated 25 simulations from the vigilant-vigilant (= state of individual 1—state of individual 2) state, 25 from vigilant-feeding, 25 from feeding-vigilant and 25 from feeding-feeding. Once initiated, a random number from the exponential distribution fit to that individual's vigilant and feeding durations determined at what point the individual switched its state. For example, let us say we want to simulate data for individuals A and B starting with the state feeding-vigilance. So, the value of behavior of A at time 0 is 0 and for B at time 0 is 1. We pick a random number from the exponential distribution fit to A's feeding durations; let us say it is 6.5. Then, the value for the behavior of A remains 0 until time = 6.5s and then switches to 1. We now pick a random number from the exponential distribution fit to A's vigilant durations; let us say it is 3.3. Then A retains the value of 1 from time = 6.5s until time = (6.5+3.3) = 9.8s and then switches to 0. This continues until the bout duration (determined previously; see above) is reached. The same is done for B.

All behavioral bouts obtained were lowpass filtered with a filter of 2.78 Hz (using MATLAB's 'lowpass' function from the 'signal processing toolbox'), and Hilbert transform applied (using MATLAB's 'hilbert' function from the 'signal processing toolbox') to obtain time series of phase angles. The Kuramoto model was fit to the simulated bouts, and coupling constant and critical coupling constant values were obtained using the same procedure described previously.

## The non-periodic Kuramoto model

The Kuramoto model assumes that all oscillators are inherently periodic. However, Sarfati et al. [74] recently showed that fireflies that do not inherently periodically flash can

synchronize their flashing. Inspired by this phenomenon and the fact that we do not expect individual marmosets to demonstrate perfect periodicity in head oscillations, we came up with a modified form of the Kuramoto model, which does not assume that the oscillators are inherently periodic. We propose the equations of this non-periodic Kuramoto model for the marmoset pairs in our data by modifying Eqs (3) and (4) as follows:

$$\frac{d\theta_1}{dt} = \frac{2\pi}{\tau_{V,1} + \tau_{F,1}} + \frac{K}{2}\sin\left(\theta_2 - \theta_1\right)$$

$$\frac{d\theta_2}{dt} = \frac{2\pi}{\tau_{V,2} + \tau_{F,2}} + \frac{K}{2}\sin\left(\theta_1 - \theta_2\right)$$

$$\tau_{p,i}(t) = \begin{cases} \tau_{p,i}(t-1), -\frac{\pi}{2} < \theta_i(t-1) \leq \frac{\pi}{2} \text{ and } -\frac{\pi}{2} < \theta_i(t) \leq \frac{\pi}{2} \\ \tau_{p,i}(t-1), \left[\theta_i(t-1) \leq -\frac{\pi}{2} \text{ or } \theta_i(t-1) > \frac{\pi}{2}\right] \text{ and } \left[\theta_i(t) \leq -\frac{\pi}{2} \text{ or } \theta_i(t) > \frac{\pi}{2}\right] \\ X_{p,i,t} \sim Exp\left(\mu_{p,i}\right), \text{ otherwise (including when } t = 0) \end{cases}$$

$$where\ p \in \{V, F\}, i \in \{1, 2\}, \theta_i \in [-\pi, \pi] \qquad (12)$$

Here, subscript V stands for the vigilant state and F for the feeding state. $\mu_{p,i}$ is the mean of the exponential distribution fit to durations of state p for individual i, and $X_{p,i,t}$ is a random number drawn from that exponential distribution at time t. The random variable $\tau_{p,i}$, which represents the time period for which individual i would remain in state p, makes the non-periodic Kuramoto model different from the classic model. In the non-periodic model, the natural frequency of an individual resets to a random number obtained from the empirical data whenever the individual undergoes a transition in the state. Transitions occur at $\theta_i = \pi/2$ and $\theta_i = -\pi/2$.

## Fitting the non-periodic Kuramoto model to the data

For all individuals in the data, we determined the mean of the exponential distribution fit to vigilant durations ($\mu_{V,i}$) and feeding durations ($\mu_{F,i}$). Then, for every bout in the experimental data, we determined the duration and the initial phases of the individuals ($\theta_1(0)$ and $\theta_2(0)$). Using these initial phases as the starting point, we performed numerical simulations of Eq (11) employing Euler's method and performing a parameter sweep of K between values -5 Hz and 0 Hz and step size of 0.1 Hz (equating to 50 values of K). For each value of K, we performed 10 numerical simulations with different random number seeds, which slightly varied the output due to the behavior of the random variable $\tau_{p,i}$. Therefore, for every behavioral bout in experimental data, we performed 500 non-periodic Kuramoto model simulations. For every simulation, we obtained the time series of phase differences ($\theta_2 - \theta_1$). Then, for each value of K we determined the mean of the angular distance between each of the 10 simulated time series of phase differences and the time series of phase differences from the actual behavioral bout. By performing the parameter sweep of K, we determined the value of K for which the angular distance was the minimum. This K-value provided the best fit of the non-periodic Kuramoto model to the data and was, therefore the estimated coupling constant for the behavioral bout.

## Comparing models

We compared the K values obtained by fitting the non-periodic Kuramoto model to the actual bouts to that of the values obtained by fitting the classic Kuramoto model to the actual and

control bouts. Further, to assess model fits, we devised a Mean Absolute Error (MAE) metric which is the mean (averaged over a behavioral bout) of the absolute differences between the observed instantaneous relative phase $\Phi(t)$ in radians and that predicted by the model. We compared MAEs of the classic and non-periodic Kuramoto models fit to the actual bouts.

## Statistical analysis

We used non-parametric tests to test our hypotheses for two reasons: (1) our sample consisted of n = 14 individuals, but most tests compared pairs which were 7 and (2) in many cases, we were dealing with ratios and exponential distributions where we couldn't assume that the distribution underlying the outcome variable was normal. For comparing 2 groups, we applied Wilcoxon signed-rank tests (using MATLAB's 'signrank' function from the 'statistics and machine learning toolbox') and for more than 2 groups, we used Friedman's test (on RStudio using 'FriedmanTest' from the package 'PMCMRPlus') as the global statistical test followed by post-hoc Nemenyi test (on RStudio using 'frdAllPairsNemenyiTest' from 'PMCMRPlus') to find which groups differed significantly if the Friedman test detected a significant difference. To compare linear regression lines, we performed non-parametric analysis of covariance (non-parametric ANCOVA on RStudio using 'sm.ancova' from the 'sm' package). Further, to check for effects of the nature of the session (inside with 1 bowl/ inside with 2 bowls/ outside with 1 bowl/ outside with 2 bowls), we compared linear regression lines of the 4 types of sessions for the actual bouts again using non-parametric ANCOVA.

To investigate the effect of the location of the marmosets (inside/outside) and the number of bowls (1/2) on the coupling constant, we first checked if the coupling constant values obtained from bouts followed a normal distribution using a qq-plot and Kolmogorov-Smirnov test. Because the values followed a normal distribution (p<0.001, Kolmogorov-Smirnov test for normality), we fit a Gaussian linear mixed-effect model with location, number of bowls and the interaction between them as fixed effects and the pair id (according to S3 Table) as the random intercept. Further, using ANOVA, we compared our model to the null model (only including the random intercept).

## Supporting information

**S1 Table. Results from linear mixed-effect model fit.** ANOVA table for the fixed effects in the model K ∼ 1 + Location + Bowls + Location:Bowls + (1|Group).
(XLSX)

**S2 Table. Ethogram with definitions for all behaviors/categories coded from video material.** Table taken from Brügger et al. 2023 [68].
(XLSX)

**S3 Table. Individuals included in study.** List of all individuals involved in the study detailing their age, status and time as well as order of testing. i = inside; o = outside; 1 = one food bowl; 2 = two food bowls
(XLSX)

**S1 Fig. Distributions of vigilance and feeding durations of marmosets when alone.** Plots depict histograms of the probability density functions (PDF) of feeding (**A**) and vigilance (**B**) durations across n = 14 individuals when they were alone, and negative exponential, log-normal and normal distributions fit to the histograms.
(TIF)

**S2 Fig. Contributions of vigilant and feeding durations to the increase in time period seen when marmosets are together.** Plots compare the fit parameters of vigilance (**A**) and feeding durations (**B**) in the alone (Al) and together (To) conditions. Each point is an individual (n = 14 individuals). Individuals belonging to the same group are connected by lines. \*\*p<0.01, two-sided Wilcoxon signed-rank test.
(TIF)

**S3 Fig. Distributions of bout durations.** Histograms for all bouts in the data (**A**), bouts for which the Kuramoto model could be fit (**B**), and analyzed bouts (**C**).
(TIF)

**S4 Fig. Effect of interaction between location and number of bowls on the coupling constant (K).** Each point is a behavioral bout (n = 38 bouts). Dots with error bars depict the estimated means and 95% confidence intervals. \*p<0.05, NS = p>0.05, post-hoc comparisons of estimated means (Welch's t-test).
(TIFF)

**S5 Fig. Overview of experimental conditions.** This schematic shows the view from within home enclosures. Individuals experienced either of these conditions (inside 1 bowl, inside 2 bowl, outside 1 bowl, outside 2 bowls) in a randomized order (see S3 Table) in their respective home enclosures. Animal were fed from either 1 or 2 feeding bowls located inside a feeding basket on the front of enclosures. During the whole experimental duration, animals were able to move freely within the respective location. Cameras were place both outside and inside of home enclosures (not shown). Video coding was done frame by frame according to definitions from S2 Table and specifically included all looking behavior over arms reach and not at a conspecific in any location of the enclosure as well as feeding behavior. For data analysis the datasets were restricted to either the "together" condition (blue bowls) where animals were situated together on the feeding basket or "alone" condition (yellow bowls) where animals were situated alone on the feeding basket. Note that even though "together" and "alone" conditions are only shown for two out of four possible conditions animals' data was present for all four conditions. Note that the marmosets are depicted disproportionally large relative to the size of the overall enclosures.
(TIF)

## Acknowledgments

We thank A. Götschi for data coding, A. Godard for help with interobserver reliabilities and G. Bazzell for animal caretaking as well as support during data collection.

## Author Contributions

**Conceptualization:** Nikhil Phaniraj, Rahel K. Brügger, Judith M. Burkart.

**Data curation:** Rahel K. Brügger.

**Formal analysis:** Nikhil Phaniraj, Rahel K. Brügger.

**Funding acquisition:** Judith M. Burkart.

**Investigation:** Nikhil Phaniraj.

**Methodology:** Nikhil Phaniraj.

**Project administration:** Judith M. Burkart.

**Resources:** Judith M. Burkart.

**Supervision:** Judith M. Burkart.

**Validation:** Rahel K. Brügger.

**Visualization:** Nikhil Phaniraj, Rahel K. Brügger.

**Writing – original draft:** Nikhil Phaniraj, Rahel K. Brügger.

**Writing – review & editing:** Judith M. Burkart.

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
