## [Decision Letter · Decision Letter 0]

25 Nov 2023

Dear Mr. Phaniraj,

Thank you very much for submitting your manuscript "Marmosets mutually compensate for differences in rhythms when coordinating vigilance" for consideration at PLOS Computational Biology.

As with all papers reviewed by the journal, your manuscript was reviewed by members of the editorial board and by several independent reviewers. As you can see from the reviewers comments below, there is support for your work but a number of issues have been raised. We would like to invite the resubmission of a significantly-revised version that takes into account the reviewers' comments.

We cannot make any decision about publication until we have seen the revised manuscript and your response to the reviewers' comments. Your revised manuscript is also likely to be sent to reviewers for further evaluation.

Sincerely,

Ralph Simon

Guest Editor

PLOS Computational Biology

Zhaolei Zhang

Section Editor

PLOS Computational Biology

Reviewer's Responses to Questions

**Comments to the Authors:**

Reviewer #1: This paper presents models and data to determine how animals coordinate their vigilance. The paper shows evidence that marmosets in pairs coordinated their vigilance and that this was achieved through mutual adaptation as the rhythm of each individual differed when they were alone.

The paper makes both empirical and theoretical contributions, which is quite rare and highly relevant to a computational journal. I can see the value of adapting the classic Kuramoto model to non-periodic cases. However, I am not able to determine whether this was done correctly as the mathematics eluded me. Most of my comments below are thus directed at the biology. Many comments probably underscore my lack of knowledge about this type of modelling. At the same time, if the paper is to reach a behavioural ecologist like myself, some changes are needed. Nevertheless, I think it is fascinating to use coupled oscillators to investigate synchrony in vigilance. This is much better than the simple approach used in the past, which looks at group level patterns but eschews individual adjustments.

Line 86 : Should this not be anti-phase synchronization?

Line 88 : The evidence for coordination is actually quite good for sentinel-like systems. Evidence for cooperation is less common in non-sentinel systems when all individuals are actively foraging. It typically involves small groups like families in cranes or pairs in coral fish. Perhaps expand on this a little.

Line 106: Are there alternatives to the Kuramoto model? Please justify this choice a little more. Also, explain what it means when oscillators are coupled? Perhaps give some examples when this sort of coupling occurs. I have a hard time imagining how oscillators can be coupled for non-living oscillators and evolve to a steady state.

Line 120 : Many models predict regular vigilance, meaning that the duration of vigilance bouts is centred on a particular value rather than taking on a wide range of values. This would lead to quite periodic oscillations I would think. What is the evidence that marmoset vigilance is not regular?

Line 158 : This could be a good place to provide details about the feeding apparatus so that the reader can understand how vigilance and feeding are mutually exclusive.

Line 160 : I doubt that potential predators are present near the enclosures but I suppose marmosets are not totally aware of this.

Line 166 : Did you watch the videos frame by frame to determine the beginning and ending of behavioural bouts and if so what was the frame rate?

Line 196 : I was lost here with this sort of transformation. It was not totally clear to me why this was needed.

Line 199 : I was not sure what phase angles mean practically.

Line 203 : As mentioned earlier, regular vigilance is also possible at least in other species. The Poisson model assumes that the rate of interruptions is constant, thus yielding a negative exponential distribution of duration lengths. Have you tried a normal distribution, which would fit with the regular model, or a log-normal, which might be appropriate under certain circumstances? At least make the modelling choice more transparent.

Line 206 : I was lost here again. What is a ratio of fit parameters? Please define the fit parameter and then how this ratio was calculated. Also, explain why this is needed and how to interpret their values.

Line 222 : Please explain the terminology for this sort of model. I doubt many readers will know the meaning of terms like phase and natural frequency.

Line 244 : With so many data points coming from the same individuals, there is a need I think to control for auto-correlation, no? This looks like a time series to me with all the trappings of correlated data.

Line 400 : Judging from the data distribution for both vigilance and feeding duration in figure 3, it appears that the fit to the exponential distribution must be rather poor as the distributions have humps. Like mentioned earlier, have you tried the normal or log-normal distributions? An overview of the different types of distributions that can be fitted with vigilance data please see Biology 11, 1769.

Line 404: Values closer to 1 suggest synchrony. This is not clear to me. This certainly means that the duration of vigilance bouts is now more similar when in pairs but does it really say anything about when those vigilance bouts occur for each individual? These bouts could still be totally asynchronized between pair members, no?

Line 412 : In figure 3d, should you not compare the results controlling for pair id like in figure 3c? In the together condition, the data points are not independent.

Line 415 : I am not entirely sure I understand the logic behind the argument that looking at mean time period of oscillations can tell us something about synchrony or coordination. Please elaborate.

Line 436 : Much of what is shown here was already established in the methods section. I think this section should be moved there.

Line 437 : Could you not simply run a standard time series analysis to determine if oscillations were periodic when the individuals were alone? If this is the case, we would not need the non-periodic Kuramoto model.

Figure 4 : Please explain which angles are compatible with synchronization or coordination.

Line 521 : I am not sure how we can conclude that the oscillations were non-periodic. In the results, I saw that the K constant was not the same for the periodic and non-periodic models. I cannot see how this can be used to assess model fit. Is there a way to show with a p-value which model best fitted the data?

Line 533 : As stated earlier, I am not fully convinced that the exponential is the best fit to the data. Also, it would be nice to discuss the rich literature on fitting functions to the distribution of vigilance duration. In many cases, the exponential was a poorer fit than other distributions. Exponential-like distributions are expected when individuals face stalking predators that can approach surreptitiously when foragers are not vigilant. Would this apply to marmosets? Just to be clear, it is the probability of changing state that is independent of the duration of the preceding state.

Line 542: We would expect vigilance to go down when animals are in pairs compared to when they are alone and also to spend more time feeding. Therefore, I was not expecting the results that marmosets were spending more time in each behavioural state when together. This should be true only for feeding duration. From figure 3a, one individual was more vigilant when in pair than alone. Was it a common finding? It would be nice to see what happens to mean duration of vigilance and feeding when alone or in pairs to judge the group-size effect. If the individuals converge in vigilance time, we would expect that each one could only feed for 50% of the time if coordinated, perhaps a value lower than when some marmosets were alone. Did this happen in these pairs and if so it would indicate a cost of coordination.

Line 544: A slowing down of oscillations when in pairs does not require monitoring the state of the companion. As per the group-size effect on vigilance and feeding, either a reduction in vigilance time and/or an increase in feeding time would be sufficient.

Line 555: Please provide references to the safety argument. It gives the impression that the authors are the first to have thought of this.

Line 556: The ethological argument is a little sketchy. There is no reason a priori why individuals should synchronize or coordinate their vigilance. Maintaining independent vigilance would still provide benefits when individuals are in groups by reducing the time spent on vigilance and by increasing the time spent feeding. Some have argued that there is a benefit to synchronize vigilance (i.e. phase synchrony) if predators are more likely to target laggards (those less vigilant at the time of an attack). In this case, individuals want to avoid being the only one non-vigilant when others are vigilant. Therefore, they all tend to adopt the same state at the same time. This can lead to long periods when no one is vigilant. Coordination brings the most benefit at the group level by minimizing periods where no one is vigilant, but this comes at the cost of monitoring neighbours and ensuring that cheating does not occur.

Line 572: But why is the coupling stronger when feeding from two bowls? The lack of space would suggest a weaker coupling when feeding from two bowls. Can they monitor one another more easily in on situation than the other?

Line 584: How do we know that each individual changed its behaviour to coordinate with the other rather than one individual showing flexibility but not the other one? Did I miss the evidence for mutual adaptation rather than unilateral adaptation?

Line 624: As mentioned earlier, I was not sure how it is possible to determine how one model, periodic or not, provides a better fit than the other.

Reviewer #2: The manuscript “Marmosets mutually compensate for differences in rhythms when coordinating vigilance” aims at modelling the vigilance behaviour of pairs of captive marmosets when feeding. The authors model the monkeys’ behaviour as (inherently aperiodic) coupled oscillators. The authors' main result is that marmosets take turns in their vigilance/feeding behaviours in a flexible way (i.e., variable coupling strength).

The study is very interesting, well-designed, and of potential interest to a wide audience. In addition, the authors extend the Kuramoto model to the case of non-periodic oscillators, which could be a very useful approach for many researchers. As I am not a modeller, however, I will refrain from commenting specifically on the modelling aspects of the manuscript.

Despite the positive points, I am uncertain about some methodological aspects, and—more in general—I think that the manuscript could be still improved in terms of clarity. In the following, I detail my concerns and questions to the authors.

Major points:

- The kind of behaviour the authors focus on is not clear to me. More specifically:

A) In the introduction, the authors broadly describe behaviours related to the sentinel system, but at line 120 they put the focus on “head oscillations” without having explained why. This should be better motivated and integrated with the previous part of the introduction.

B) While the focus here is on head oscillations, head movements were either not quantified or their quantification was not described, as the only coded relevant behaviour is “vigilance”. Could the authors explain better why they refer to head oscillations?

C) Did the authors collect any measure on the direction of the head (or eyes) movements?

-Lines 123-126: Please explain why you expect marmosets not to follow simple interaction rules during this behaviour.

-I assume the animals were born in captivity. Because of this, and since they are adult animals, I would expect them to know that no harm and no predators will come for them while they feed.

A) Can the authors (maybe in the discussion) speculate on whether they think the observed results would hold in wild animals, why yes or why not?

B) In line 160, the authors talk about a distinction between inside and outside as different risk zones based on the presence of “potential predators”. Shouldn’t the animal be habituated to both areas though, thus knowing no predators will come in either?

C) Related to the previous aspect, and because no direct measure of head/eyes movements was taken, I am wondering how we can be sure that the observed behaviour reflects vigilance rather than simpler taking turns to eat from the bowl. Could the authors elaborate on this?

- I am not sure I understand the control condition of the animal being alone in the feeding basket. Is it necessary for the companion animal to also be in the basket to exert vigilance on the surroundings? I would imagine that a nearby position would be equally effective. Maybe a picture of the real set-up, or a figure would be helpful here (see also my comments below).

- In the methods (lines 201 and following), the authors describe how they modelled vigilance in alone and together conditions. What about the other conditions? I would assume that also inside/outside should be relevant here.

- Related to the previous point, the other conditions (1 vs. 2 bowls and inside vs. outside) are mostly reported in the supplementary materials. I suggest making this more clear and explicit in the main text.

Minor:

-Line 45-46. The construction of the sentence especially after the colon is unclear. Please rephrase.

-Line 55-56: What do the authors mean by “non-human animals with more complex cognitive abilities”? Cognitive abilities were not explored up to that point in the manuscript, nor specific animal groups or species with “lesser” abilities. Please clarify.

-Lines 59-60: “are then confronted with novel regular and irregular sequences (“anisochrony detection”) that they are expected to generalise the patterns of regularity too” – something is o

---

## [Decision Letter · Decision Letter 1]

24 Apr 2024

Dear Mr. Phaniraj,

We are pleased to inform you that your manuscript 'Marmosets mutually compensate for differences in rhythms when coordinating vigilance' has been provisionally accepted for publication in PLOS Computational Biology.

Please note that **all data** and related metadata underlying reported findings **should be deposited in** appropriate **public data repositories**, unless already provided as part of the submitted article.

Please also note that your manuscript will not be scheduled for publication until you have made the required changes, so a swift response is appreciated.

**IMPORTANT:** The editorial review process is now complete. PLOS will only permit corrections to spelling, formatting or significant scientific errors from this point onwards. Requests for major changes, or any which affect the scientific understanding of your work, will cause delays to the publication date of your manuscript.

Best regards,

Ralph Simon

Guest Editor

PLOS Computational Biology

Zhaolei Zhang

Section Editor

PLOS Computational Biology

Reviewer's Responses to Questions

**Comments to the Authors:**

Reviewer #1: Thank you for considering my comments. I agree that the revised version is much better now. I have no further comments.

**Have the authors made all data and (if applicable) computational code underlying the findings in their manuscript fully available?**

Reviewer #1: Yes

PLOS authors have the option to publish the peer review history of their article (what does this mean?). If published, this will include your full peer review and any attached files.

Reviewer #1: No

---

## [Editor Report · Acceptance letter]

7 May 2024

PCOMPBIOL-D-23-01573R1 

Marmosets mutually compensate for differences in rhythms when coordinating vigilance

Dear Dr Phaniraj,

I am pleased to inform you that your manuscript has been formally accepted for publication in PLOS Computational Biology. Your manuscript is now with our production department and you will be notified of the publication date in due course.

With kind regards,

Anita Estes
